# Non-Indigenous Freshwater Fishes as Indicators of Ecological Quality in Running Waters

**Christos Gkenas** [1,*] **, Leonidas Vardakas** [2] **and Nicholas Koutsikos** [2]

1 MARE—Centro de Ciências do Mar e do Ambiente/ARNET—Aquatic Research Network, Faculdade de Ciências, Universidade de Lisboa, 1749-016 Lisbon, Portugal
2 Hellenic Centre for Marine Research, Institute of Marine Biological Resources and Inland Waters, 19013 Anavissos, Attica, Greece; lvard@hcmr.gr (L.V.); nkoutsik@hcmr.gr (N.K.)
* Correspondence: cgkenas@ciencias.ulisboa.pt; Tel.: +351-217-500-000 (ext. 28182)

**Abstract:** The European Union Water Framework Directive (WFD) is a comprehensive initiative guiding river basin water management, addressing pressures such as pollution from diffuse and point sources, and hydromorphological alterations. However, pressures that can deteriorate the ecological quality of water bodies through biotic changes (i.e., by the introduction of non-indigenous species) are not rigorously addressed by the WFD. This study explores associations between conventional ecological quality indices based on physicochemical and biotic quality elements (fish and macroinvertebrates) against the presence and densities of non-indigenous freshwater fish species (NIFS) in lotic ecosystems of Greece, aiming to unravel the potential usage of NIFS in future ecological assessments. The dataset comprises 277 samplings at 115 sites, covering physicochemical and biotic (fish and macroinvertebrate) quality indices, and anthropogenic pressure variables. Based on our findings, the occurrence and densities of four NIFS (*Lepomis gibbosus*, *Pseudorasbora parva*, *Gambusia holbrooki*, and *Carassius gibelio*) were highly associated with the ecological quality assessments of the applied indices. Higher occurrences and densities of these NIFS were related to samplings of lower ecological quality. In addition, NIFS exhibited a positive association with anthropogenic pressures, likely due to their adaptability to less optimal environmental conditions or higher tolerance to pollution and other stressors. Our findings underscore the need for a paradigm shift in ecological quality assessments, emphasizing the use of NIFS either as a potential indicator of ecosystem health or as a pressure that deteriorates ecological quality.

**Keywords:** water framework directive; ecological quality; metrics; alien fish; river; Greece

## 1. Introduction

The EU Water Framework Directive (WFD) is a visionary initiative by the European Union aiming to establish wide-ranging guidelines and strategies for river basin water management. The directive sets ambitious goals related to water quality, alongside addressing emerging challenges such as climate change impact, urbanization, and evolving pollution sources [1]. The fundamental objective of the WFD is to achieve good ecological status in all surface water bodies (i.e., rivers, lakes, transitional waters, and coastal waters). The directive necessitates member states of the EU to classify the ecological status of each water body, ranging from undisturbed conditions (reflecting high ecological quality) to highly disturbed conditions (indicating bad ecological quality). In water bodies that fail to achieve at least good ecological quality (i.e., slight variation from undisturbed conditions), mitigation measures should be enforced to reverse their degraded condition. To meet the WFD requirements, member states are obliged to monitor and assess the ecological quality of all surface water bodies, considering physicochemical, hydromorphological, and biological quality elements.

Non-indigenous species have been consistently associated with habitat degradation, primarily through the alteration of the structural and functional aspects of invaded ecosystems, leading to ecological imbalances and biodiversity loss. The impact of these invasions extends beyond ecological consequences, affecting economic activities and various ecosystem services [2]. Despite the high association of non-indigenous species with degraded habitats [3], the WFD does not explicitly refer to these species either as a pressure or as a potential indicator of ecosystem health. In fact, the WFD merely acknowledges the potential threats posed by these species within the objectives outlined in the directive's annexes, categorizing them as "other pressures" [4]. Furthermore, non-indigenous species are often excluded as biological components in quality indices used to assess the overall ecological quality of specific water bodies under the WFD [5]. In most cases, biotic indices are typically designed to evaluate ecological health including solely native species, while overlooking the impact of non-indigenous species [6,7]. This gap in both acknowledgment and assessment can hinder a holistic understanding of an ecosystem's health.

The presence, richness, density, and biomass of non-indigenous freshwater fish species (NIFS) have been frequently integrated as metrics in various indices of biotic integrity to assess ecosystem health in streams and rivers in the USA (see references in [8]). This acknowledges the significance of NIFS in shaping the dynamics of aquatic ecosystems, and their inclusion in ecological assessments serves as an important indicator of their impact on native aquatic communities and overall ecological equilibrium. In contrast, in Europe, only a limited number of countries include data on non-indigenous species in their ecological quality assessments, and even fewer have specific monitoring programs targeting invasive alien species [9].

Within this study, our aim was to uncover associations between the ecological quality indices applied in the framework of the Greek National Water Monitoring Programme in the period of 2012–2015 against the presence and abundance of NIFS. Specifically, we explore the relationships of physicochemical and biotic (macroinvertebrate and fish) quality elements and anthropogenic pressures with the presence and abundance of NIFS aiming to identify their potential usage in future ecological assessments.

## 2. Materials and Methods

### 2.1. Data Acquisition

We compiled data on the physicochemical and biotic status (fish and macroinvertebrates) from 277 samplings conducted at 115 sites in lotic ecosystems in the mainland part of Greece (Figure 1). In addition, for each sampling, we collected data on species richness and density (individuals per 100 m$^2$) for both native fish and NIFS, as well as anthropogenic pressures. All data were acquired from the Greek Ministry of Environment and Energy (Special Secretariat for Water) as part of the implementation of the National Water Monitoring Programme under the WFD for the 2012–2015 period.

Data collection, hydromorphological assessments, pressure evaluations, laboratory analyses, and ecological quality classification were conducted by the Institute of Marine Biological Resources and Inland Waters (IMBRIW) of the Hellenic Centre for Marine Research (HCMR). Methodological details on field samplings and the evaluation and classification of the physicochemical, hydromorphological, and biotic status can be found in corresponding references [10–14]. Briefly, the ecological quality status of samples based on benthic macroinvertebrate communities was estimated using the HESY2 (Hellenic Evaluation System 2). The ecological quality ratio of an observed value within the existing Hellenic Evaluation System-HES [15] was compared with the expected median reference value of the same river type (R-M). This ratio measures the abundance and diversity/richness of benthic macroinvertebrates (at the family level) and their tolerance to pollution and is standardized against habitat diversity and richness [11]. The HEFI (Hellenic Fish Index) determines the ecological quality status of samples derived from site-based river ichthyofauna samples. This index, developed based on modeled reference conditions, incorporates site-specific fish samples from an extensive dataset in Greece [12]. It utilizes four trait-based metrics

concerning feeding traits (insectivorous and omnivorous), feeding habitat (benthic), and migratory behavior (potamodromy) [12]. Specifically, the metrics used in HEFI are: (1) proportional density of insectivorous larger than 100 mm, (2) proportional density of omnivorous smaller than 100 mm, (3) proportional density of benthic species smaller than 150 mm, and (4) proportional density of potamodromous fishes [12]. Although NIFS were included in HEFI's assessment as part of the species pool, they were not given a special significance and their attributes and metrics were not included. Higher values in both indices indicate better quality.

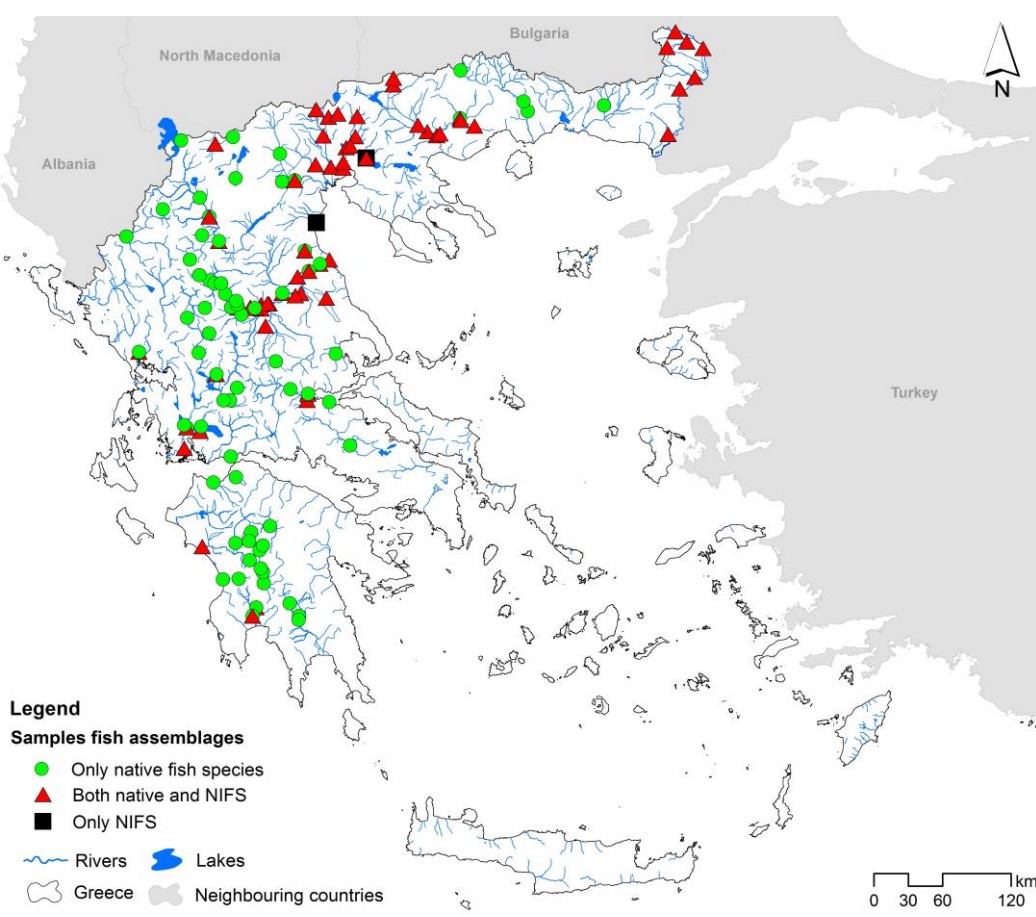

**Figure 1.** Map of Greece illustrating the rivers' sampling sites used in the present study.

The physicochemical quality of water samples was assessed by using the Nutrient Classification System (NCS). This index evaluates the chemical quality of water by considering concentrations of the nutrients: nitrate ($NO_3$), nitrite ($NO_2$), ammonium ($NH_4$), and phosphate ($PO_4$) [10]. Moreover, the concentration of dissolved oxygen (DO) is considered as an additional parameter in the assessment. In all cases, the quality status is ranked based on the WFD five-class system (i.e., "high", "good", "moderate", "poor", and "bad") [16].

The quality condition of each site, determined by the severity of each anthropogenic pressure, was assessed by IMBRIW researchers on a scale of 1 to 5, following the WFD standard (i.e., 1 = high, 2 = good, 3 = moderate, 4 = poor, and 5 = bad). Some pressures were assessed using a five-class scale (i.e., 1 = least impacted to 5 = highly impacted), while others used a three-point scale (i.e., 1, 3, and 5 or 1, 2, and 5). The twelve pressure parameters have been evaluated by researchers of IMBRIW as follows (increment score measures in parentheses): channel modification (1–5); instream habitat modification (1, 3, and 5); artificial embankment (1–5); riparian vegetation modification (1–5); barrier upstream (1, 2, and 3); barrier downstream (1, 3, and 5); barrier basin (1, 3, and 5); water abstraction (1,

3, and 5); hydropeaking (1, 2, and 5); hydrological modification (1, 3, and 5); impoundment (1–5); and pollution (1, 2, and 5).

## 2.2. Statistical Analyses

To compare fish densities across the quality indices, we utilized the Kruskal–Wallis test, followed by post hoc Dunn's multiple pairwise comparisons. This non-parametric test was selected since the assumptions of normality and homogeneity were not met, even after applying arcsine or logarithmic transformations [17]. The relationships of both NIFS and native fish species with the ecological quality indices were statistically examined using Spearman's rank correlation. Multivariate data analyses were performed on the fish data to indicate the main gradients of community variation and to detect and visualize similarities in fish samples. A preliminary detrended correspondence analysis (DCA) was applied to the dataset to determine the gradient length. The DCA revealed that the gradient lengths of the first two axes were greater than 3 standard deviation units, justifying the use of unimodal ordination techniques [18].

A canonical correspondence analysis (CCA) designed for the direct analysis of relationships between multivariate ecological data [19] was applied to ecological quality indices and anthropogenic pressures variables from 167 samplings along with the corresponding dominant native and NIFS populations. Before the CCA, fish abundances were logarithmically transformed [log $(x + 1)$] to reduce the variation in abundance. The preliminary CCA identified the collinear variables and selected a subset of them upon inspection of the variance inflation factors (VIF < 10; [18]). Eleven variables were retained as candidates for inclusion in the CCA model. Monte Carlo permutation tests (999 unrestricted permutations, $p \leq 0.05$) were used to test the significance of the axis and hence determine if the selected indices and pressure variables could explain nearly as much variation in the fish community distribution as all of the measured variables combined. Analyses were run with R software (v.4.2.2, [20]) using the packages *vegan* [21] and *ggplot2* [22].

## 3. Results

Out of a total of 277 samplings, 52 native fish species were recorded, while only five NIFS were identified. These NIFS included the pumpkinseed sunfish (*Lepomis gibbosus*), the topmouth gudgeon (*Pseudorasbora parva*), the eastern mosquitofish (*Gambusia holbrooki*), the Prussian carp (*Carassius gibelio*), and the rainbow trout (*Oncorhynchus mykiss*). In total, 159 (57.4%) of the samplings had no NIFS.

The Prussian carp demonstrated the highest occurrence among NIFS, being found at 46 sites, which accounted for 40% of the total surveyed sites. Following closely, the eastern mosquitofish occurred in 40 sites, representing 34.7% of the total, and the pumpkinseed sunfish occurred in 30 sites, making up 26% of the total sites. In contrast, the rainbow trout was captured at only one site, indicating a limited distribution within the studied area. In 79 samplings, two or more NIFS co-occurred, while in three samplings (two sites), species composition consisted exclusively of NIFS.

The presence of NIFS exhibited strong associations with physicochemical, fish, and benthic macroinvertebrate quality status. The occurrence of NIFS increased as the quality class of each index decreased (Figure 2).

Native fish species exhibited higher densities, constituting 93% of the total mean density (97.9 ind./100 $m^2$; SE = $\pm 9.8$). Native fish species densities showed substantial variability within each quality status (Figure 3; Table S1), indicating the highest values in the moderate, poor, and bad quality status for all indices. However, correlations were tested and found to be not significant ($p > 0.05$).

NIFS were present in 42.6% of the samplings, demonstrating an average density of 7.2 ind./100 $m^2$ (SE = $\pm 1.7$). NIFS densities were significantly higher (Figure 4; Table S1) in the poor quality status based on the physicochemical quality index (15.5 $\pm$ 6.9) and HEFI fish quality index (21.2 $\pm$ 7.0), and in the bad quality status for the HESY2 macroinverte-

brate quality index (77.3 ± 38.8). However, correlations were tested and found to be not significant (*p* > 0.05).

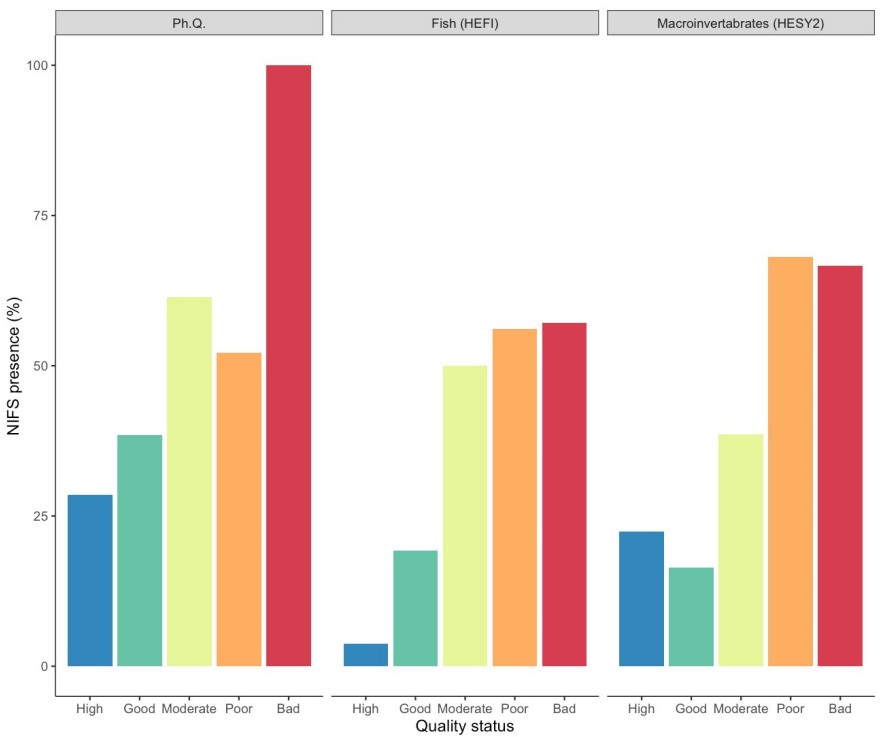

**Figure 2.** Percentage of sites with NIFS presence for each quality index and quality status. Ph. Q. based on physicochemical quality index, HESY2 based on benthic macroinvertebrate index, HEFI based on fish index.

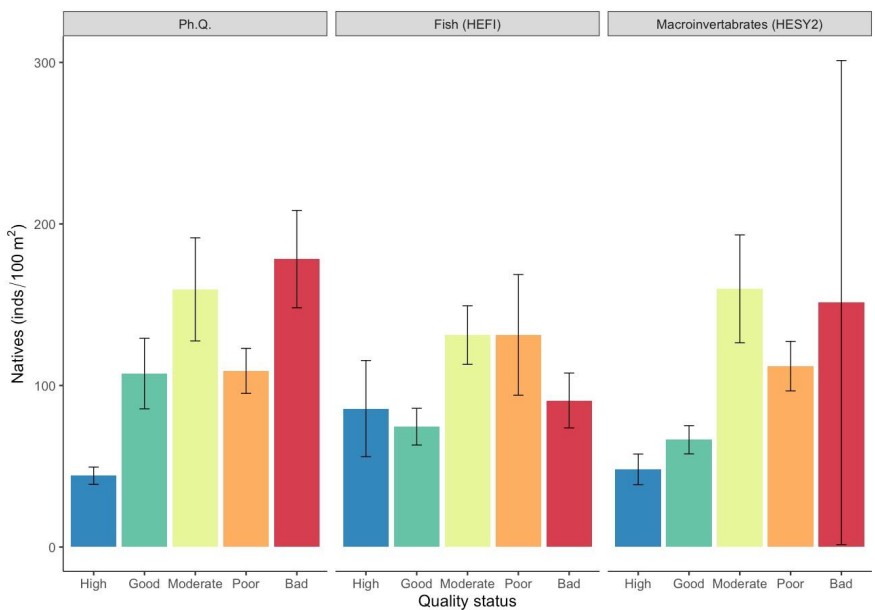

**Figure 3.** Mean density values (±SE) of native fish species in each ecological quality status based on the surveyed quality indices. (Ph. Q. = physicochemical quality index, HESY2 = benthic macroinvertebrate index, HEFI = fish index).

At the species level, the eastern mosquitofish displayed the highest densities, with a mean value of 3.5 ind./100 m$^2$ (SE = ±1.3), followed by the Prussian carp (mean density = 1.4 ind./100 m$^2$; SE = ±0.3), pumpkinseed sunfish (mean density = 1.3 ind./100 m$^2$;

SE = ±0.4), and topmouth gudgeon (mean density = 1.0 ind./100 m$^2$; SE = ±0.3). The rainbow trout, exhibited very low densities and was excluded from further analysis. The eastern mosquitofish and the Prussian carp exhibited substantially higher mean densities within the moderate, poor, and bad quality classes for all ecological quality indices (Figures 5 and 6; Table S1).

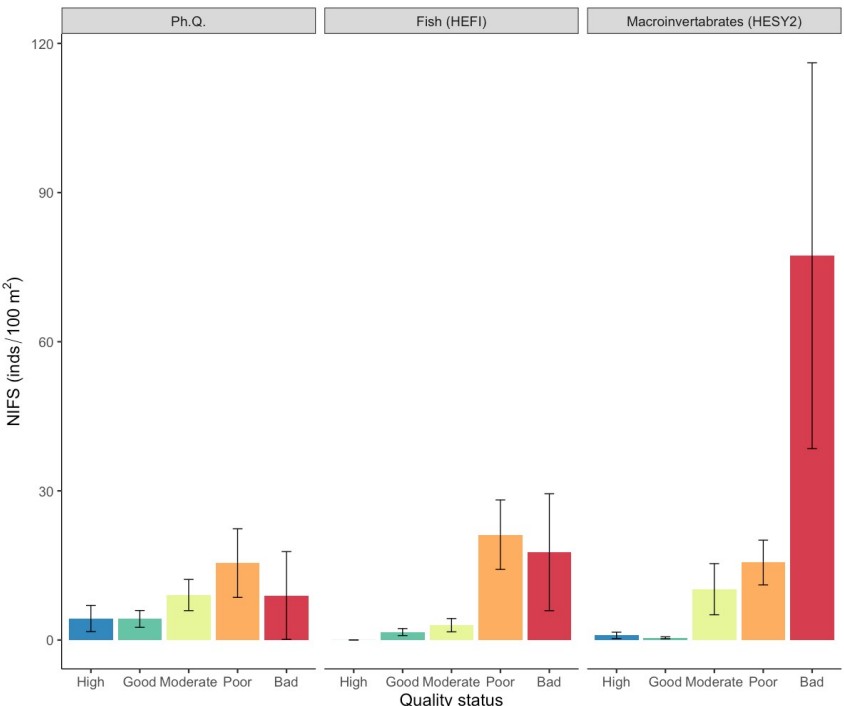

**Figure 4.** Mean density values (±SE) of NIFS in each ecological quality status based on the surveyed quality indices. (Ph. Q. = physicochemical quality index, HESY2 = benthic macroinvertebrate index, HEFI = fish index).

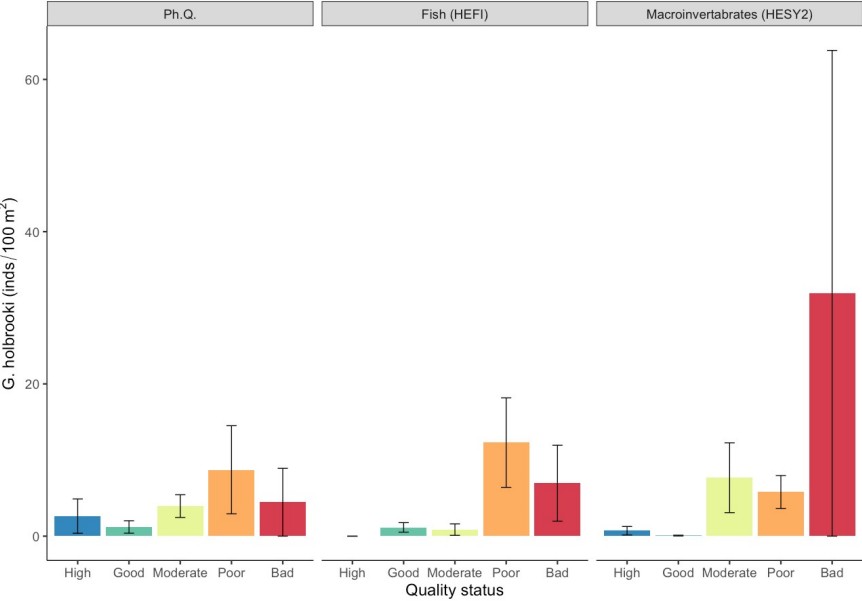

**Figure 5.** Mean density values (±SE) of *Gambusia holbrooki* in each ecological quality status based on the surveyed quality indices. (Ph. Q. = physicochemical quality index, HESY2 = benthic macroinvertebrate index, HEFI = fish index).

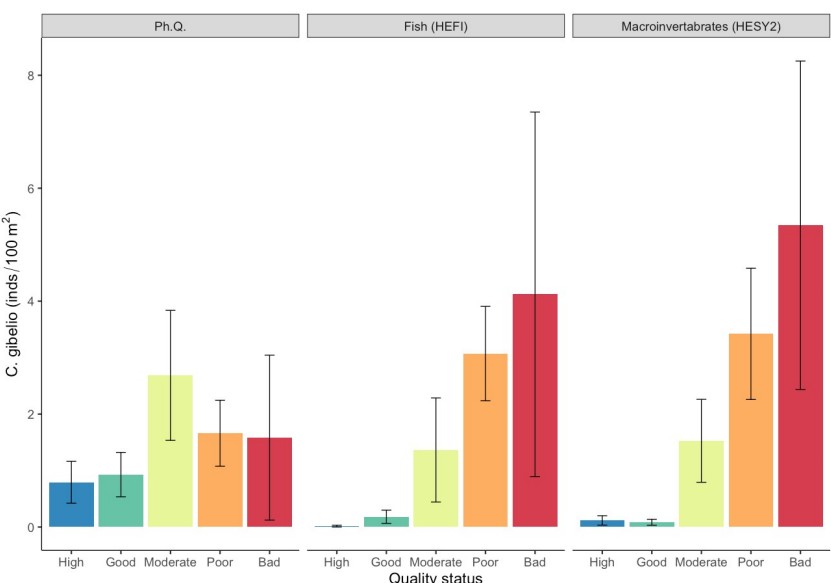

**Figure 6.** Mean density values (±SE) of *Carassius gibelio* in each ecological quality status based on the surveyed quality indices. (Ph. Q. = physicochemical quality index, HESY2 = benthic macroinvertebrate index, HEFI = fish index).

Further, the highest mean densities of pumpkinseed sunfish were observed in the bad quality class for the HESY2 macroinvertebrate quality index (36.3 ± 31.5) and the poor quality class for the HEFI fish quality index (3.9 ± 1.8), with no significant differences being evident for physicochemical quality (Figure 7; Table S1). Finally, topmouth gudgeon densities were significantly higher (Figure 8; Table S1) in the bad quality class for physicochemical quality (2.9 ± 2.9) and the HEFI fish quality index (3.5 ± 2.7), and in the poor quality class for the HESY2 macroinvertebrate quality index (3.9 ± 1.6).

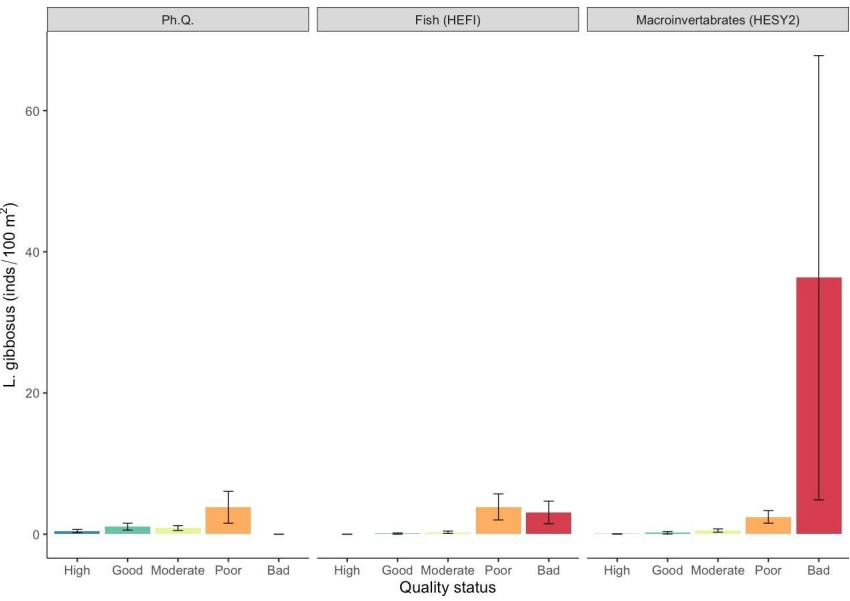

**Figure 7.** Mean density values (±SE) of *Lepomis gibbosus* in each ecological quality status based on the surveyed quality indices. (Ph. Q. = physicochemical quality index, HESY2 = benthic macroinvertebrate index, HEFI = fish index).

The ordination plot of the CCA summarized the relationships between fish densities and the set of ecological quality indices and anthropogenic pressure variables (Figure 8). CCA axes 1 and 2 (CCA1 and CCA2) jointly accounted for 30% of the variance in fish

community species composition and specific quality indices and pressure variables, respectively (Figure 9). Monte Carlo permutation showed that both axes were significant ($p < 0.001$). Among the set of quality indices and pressure variables, physicochemical, macroinvertebrate, and fish quality indices, as well as riparian vegetation and instream habitat, were mainly related to CCA1, and the other three factors—channelization, impoundment, and barriers upstream—were closely associated with CCA2. Native species were on the negative side of the first CCA axis and appeared to be strongly correlated with physicochemical, macroinvertebrate, and fish quality indices. In contrast, the Prussian carp and the topmouth gudgeon were on the positive part of the first CCA axis, positively correlated with riparian vegetation and instream habitat (Figure 6). The pumpkinseed sunfish was on the positive side of the second CCA axis, being associated with sampling sites characterized by high channelization. The eastern mosquitofish, however, showed a contrasting pattern by occupying the negative side of the second CCA axis and being associated with increased impoundment and barriers upstream (Figure 8).

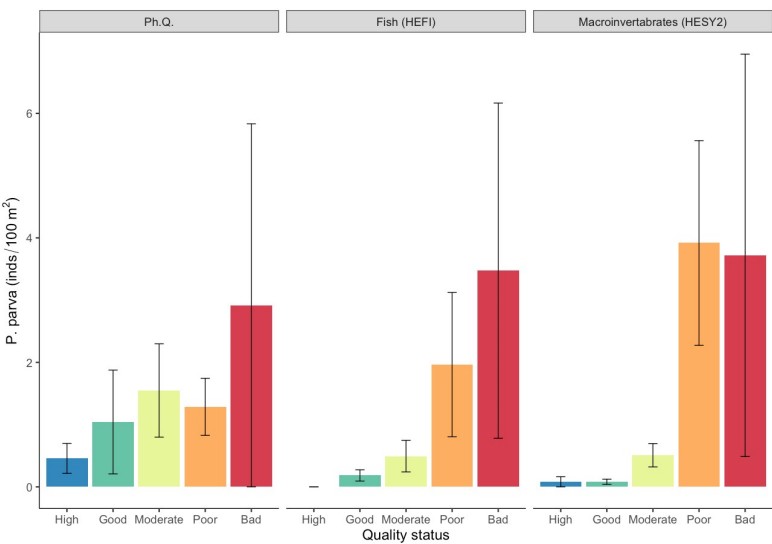

**Figure 8.** Mean density values (±SE) of *Pseudorasbora parva* in each ecological quality status based on the surveyed quality indices. (Ph. Q. = physicochemical quality index, HESY2 = benthic macroinvertebrate index, HEFI = fish index).

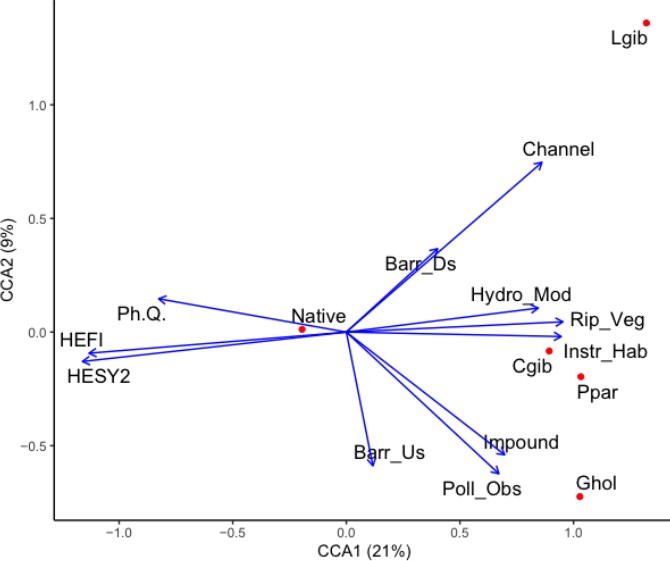

**Figure 9.** Canonical correspondence analysis biplot describing the relationship between ecological quality indices and anthropogenic pressure variables and fish species. Length and direction of arrows

indicate the relative importance and direction of change in the metrics and variables. Abbreviated names refer to the following fish names and variables: Ghol: *G. holbrooki*; Cgib: *C. gibelio*; Lgib: *L. gibbosus*; Ppar: *P. parva*; Ph. Q.: physicochemical quality index; HEFI: fish quality index; HESY2: macroinvertebrate quality index; Chann: channelization; Barr_Ds: barrier downstream; Barr_Us: barrier upstream; Hydro_Mod: hydrological modification; Rip_veg: riparian vegetation; Instr_Hab: instream habitat; Impound: impoundment; Poll_Obs: pollution observed visually on site.

## 4. Discussion

Controversy persists over whether introduced species universally harm native ecosystems, with divergent views among scientists [2]. While some assert that non-indigenous species detrimentally affect biodiversity [23–25], others argue that many species introductions, especially freshwater fish, lack ecological impact and provide significant socioeconomic advantages [26–28]. This debate, that has triggered heated disputes and mutual accusations between skeptics and invasion biologists [29–33] may be the reason for the low inclusion percentage of non-indigenous species in biotic indices. In light of the ongoing debate and limited research on non-indigenous species' impact on the ecological status of water bodies within the WFD scheme, further comprehensive studies are imperative to inform effective management strategies and policy decisions.

In this study, we explored the relationships between ecological quality indices applied within the framework of the Greek National Water Monitoring Programme against the presence and abundance of NIFS in Greek lotic ecosystems. Based on our findings, the occurrence and densities of four widespread NIFS were highly associated with the ecological quality assessments of the applied indices. Higher occurrences and densities of these NIFS were overall related to samplings of lower ecological quality. Specifically, the densities of the pumpkinseed sunfish, the topmouth gudgeon, the eastern mosquitofish, and the Prussian carp decidedly responded to the physicochemical quality index, as higher densities were found in degraded sites influenced by diffuse or distant sources of pollution, which can be reflected in physicochemical measurements [34]. In addition, these NIFS were highly responsive to instream habitat degradation, riparian vegetation alteration, and hydromorphological modification, as evidenced by the CCA analysis. Several studies have shown consistent patterns, suggesting that these four NIFS tend to become more tolerant to water pollution and habitat degradation in human-modified habitats [35–37].

Native species also showed a strong association with lower ecological quality; however, these correlations were not statistically significant. This pattern is to some extent expected, given that the dataset used to assess the ecological quality based on fish fauna comprises both reference and degraded sites, thus making native species responsive to degradation. Therefore, this pattern could potentially be influenced by factors such as water scarcity, habitat alteration, and/or the dominance of invasive species in specific sites. Additionally, the use of NIFS as indicators in significantly altered ecosystems by human activity, resulting in a mixture of native and non-native species, could be valuable for future ecological assessments. While it is well known that the ecological traits of many native species can serve as indications of anthropogenic pressure, NIFS presence and density could be also included as metrics in a quality index or utilized as distinct disturbance bioindicators due to their specific characteristics (e.g., species traits, occurrence in modified ecosystems, and absence of sampling risks) and the advantages they gain (omnipresence, high biomass and dominance, and ease of capture) within invaded ecosystems [38–40]. In addition, the four NIFS of this study showed a positive association with various anthropogenic pressures, mainly due to their adaptability to less optimal environmental conditions or their higher tolerance to pollution and other stressors. More specifically, NIFS assemblages display spatial variations, portraying distinct fish community structures that predominantly arise from the diverse habitat preferences and life-history traits exhibited by the species [41,42]. Typically, anthropogenic pressures in lotic ecosystems tend to alter river habitats to re-

semble more lacustrine environments, primarily through hydrological modifications and impoundments, or via channelization and instream habitat alterations. Consequently, these alterations tend to attract and facilitate the prevalence of non-indigenous species with limnophilic traits [42], such as the recorded NIFS in our study. Such observations align with previous research, indicating further that NIFS tend to thrive in suboptimal riverine environmental conditions, as these favor their invasion success [43,44].

According to Annex V of the WFD, a water body cannot attain a high ecological status unless its taxonomic composition and species abundance closely align with, or entirely resemble, undisturbed conditions [1]. Currently, there is a lack of consensus among member states of the EU regarding the use of non-indigenous species to determine the status of a water body and whether the mere presence of non-indigenous species should automatically hinder the attainment of high ecological status [45]. In this study, we found that NIFS were also present in samplings indicating high and good ecological quality status. This is mainly attributed to the fact that NIFS are not included as a pressure factor within ecological quality assessments, which would otherwise lead to the downgrading of the classification of a water body [45]. However, it was evident that NIFS occurrence increased in samplings with lower ecological quality. Specifically, two NIFS, namely the eastern mosquitofish and the Prussian carp, exhibited higher mean densities in moderate, poor, and bad quality classes for all quality indices investigated, indicating their association with lower ecological qualities. It is well established that high abundances of non-indigenous species signify elevated environmental pressure and a state of low ecological status [4]. Our findings emphasize the potential utility of incorporating the pumpkinseed sunfish, the topmouth gudgeon, the eastern mosquitofish, and the Prussian carp into the implementation of ecological quality assessment procedures either as an indicator of ecosystem health or as a pressure deteriorating ecological quality. While the latter limnophilic species are known to occur in degraded habitats with lower ecological status, future efforts should be devoted to addressing the presence of rheophilic species, which usually occupy undisturbed sites assessed to have high ecological status.

Incorporating non-indigenous species into ecological quality assessments presents significant challenges in designing comprehensive studies that can disentangle the complex interactions between non-indigenous species and other anthropogenic pressures. This complexity arises from the fact that the impact of invasive species may either overshadow the effects from other pressures or be masked by them [4,46–49]. It is widely acknowledged that some NIFS can alter the structure and functioning of the environment through predation, competition, hybridization, disease/parasite transmission, food web alterations, and habitat degradation [50–52]. These alterations can have cascading effects on the entire aquatic ecosystem, impacting biodiversity, ecosystem function, and services, leading to economic impact [53,54]. Therefore, NIFS can significantly influence the sensitivity of ecological quality indices, as their impact resembles other anthropogenic pressures, thereby impacting the accuracy of ecological evaluations. Researchers and environmental managers need to develop advanced indices designed to capture these specific interactions when interpreting ecological quality indices [49]. This will ensure that the accuracy of assessments remains high and that the ecological implications of non-indigenous species are adequately captured within the broader context of water quality and ecosystem health [4].

The findings of this study contribute to the growing body of research on the ecological implications of NIFS in aquatic ecosystems [4,9]. To gain a more comprehensive understanding of the role of NIFS in shaping ecosystem health, future studies should expand the scope and geographical coverage, incorporating data from diverse ecosystems (e.g., lentic environments) as well as including additional NIFS. By integrating the ecological dynamics of NIFS into the WFD, we can better address the challenges posed by these species and work towards the sustainable management of our water bodies. This will require ongoing research, monitoring, and international collaboration to address the complex ecological interactions and impact of NIFS, ultimately contributing to the long-term health and resilience of aquatic ecosystems. In conclusion, our study emphasizes the need for

a paradigm shift in the assessment of aquatic ecosystems, incorporating NIFS as integral components of the evaluation process. The presence and abundance of NIFS should be recognized as vital indicators of ecosystem health. Their inclusion in ecological assessments can lead to a more comprehensive understanding of the intricate interplay between native and non-indigenous species and inform effective conservation and management strategies.

**Supplementary Materials:** The following supporting information can be downloaded at https://www.mdpi.com/article/10.3390/d16010009/s1, Table S1: Results of Kruskal–Wallis tests on variation in the fish abundances among quality elements for native and non-native species.

**Author Contributions:** C.G.: methodology, format analysis, writing—original draft preparation. L.V. and N.K.: conceptualization, methodology, writing—review and editing. All authors have read and agreed to the published version of the manuscript.

**Funding:** Financial support was provided by Fundação para a Ciência e a Tecnologia (FCT) through the projects UIDB/04292/2020 and UIDP/04292/2020 awarded to MARE and through project LA/P/0069/2020 granted to the Associate Laboratory ARNET. C. Gkenas (DL57/2016/CP1479/CT0036) was supported by individual contracts from FCT.

**Institutional Review Board Statement:** Not applicable.

**Data Availability Statement:** The data presented in this study are available from the corresponding author on request. The data are not publicly available due to the data were acquired from the Greek Ministry of Environment and Energy (Special Secretariat for Water) as part of the implementation of the National Water Monitoring Programme under the WFD for the 2012–2015 period and need to be contacted for further use.

**Acknowledgments:** We thank S. Tasoglou from the Greek Ministry of Environment and Energy (Special Secretariat for Water), who provided us with the data of the implementation of the National Water Monitoring Programme under the WFD for the 2012–2015 period.

**Conflicts of Interest:** The authors declare no conflict of interest.

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
