# Peer review of "Non-Indigenous Freshwater Fishes as Indicators of Ecological Quality in Running Waters"

_diversity, doi:10.3390/d16010009_

Round 1

Reviewer 1 Report

Comments and Suggestions for Authors

The manuscript „Non-indigenous freshwater fishes as indicators of ecological quality in running waters“ aims to uncover associations between the ecological quality indices applied in the EU Water Framework Directive and the abundance of non-indigenous fish species (NIFS). Specifically, authors assessed the relationships of physicochemical and biotic quality elements and anthropogenic pressures with the presence and abundance of four NIFS aiming to identify their potential usage in future ecological assessments. In the conclusions, authors emphasized the potential efficacy of incorporating NIFS into the implementation of ecological quality assessment procedures either as indicator of ecosystems health or as a pressure deteriorating ecological quality. Overall, paper is well written and easy to understand. The manuscript aims are clear and well presented.

General notes.

However, there are some concerns that gives some doubts on the overall conclusion. Authors provides very much generalized opinion on the need to include NIFS into the implementation of ecological quality assessment procedures only from the investigation of four non-indigenous species (Lepomis gibbosus, Pseudorasbora parva, Gambusia holbrooki, and Carassius gibelio). Furthermore, all these non-native species are known to prefer altered river habitats that resembles more lacustrine environments with low current, well expressed vegetation and eutrophic trends. In general, there are many non-indigeneous fish species which possess completely opposite habitat requirements. A good example could be Oncorhynchus mykiss, which was excluded from this analysis by the authors or Sander lucioperca, Salmo salar, Neogobius fluviatilis. Such species would more represent habitats of a good ecological quality as they prefer running waters rich in oxygen and hard substrates. There are also many indifferent NIFS, which would not present neither bad quality habitats, nor good state habitats. Overall, my suggestion would be to avoid such generalizing statement, that all NIFS could be incorporated as good indicators for assessment of bad water ecological status, but rather talk on possibility to include the particularly assessed four NIFS species into the implementation of ecological quality assessment procedures as indicators of degraded habitats. Overall, it is well known that these four assessed NIFS (Lepomis gibbosus, Pseudorasbora parva, Gambusia holbrooki, and Carassius gibelio) prefers degraded habitats. Thus, particularly these species incorporation into EU Water Framework Directive as biotic indices for ecological health evaluation could be promising.

Specific notes

Discussion chapter:

Line 229: I suggest pointing out the names of particularly assessed non-indigenous fish species rather then to generalized as NIFS. I am certain, that the findings of provided relation trend was very highly depended on the particularly investigated non-indigenous fish species.

Line 239: Similarly as above suggesting to change NIFS into four particularly fish species Lepomis gibbosus, Pseudorasbora parva, Gambusia holbrooki, and Carassius gibelio.

Line 267: Similarly as above suggesting to change NIFS into four particularly fish species Lepomis gibbosus, Pseudorasbora parva, Gambusia holbrooki, and Carassius gibelio.

Author Response

Reply to Reviewer #1:

The manuscript “Non-indigenous freshwater fishes as indicators of ecological quality in running waters” aims to uncover associations between the ecological quality indices applied in the EU Water Framework Directive and the abundance of non-indigenous fish species (NIFS). Specifically, authors assessed the relationships of physicochemical and biotic quality elements and anthropogenic pressures with the presence and abundance of four NIFS aiming to identify their potential usage in future ecological assessments. In the conclusions, authors emphasized the potential efficacy of incorporating NIFS into the implementation of ecological quality assessment procedures either as indicator of ecosystems health or as a pressure deteriorating ecological quality. Overall, paper is well written and easy to understand. The manuscript aims are clear and well presented.

General notes:

However, there are some concerns that gives some doubts on the overall conclusion. Authors provides very much generalized opinion on the need to include NIFS into the implementation of ecological quality assessment procedures only from the investigation of four non-indigenous species (Lepomis gibbosus, Pseudorasbora parva, Gambusia holbrooki, and Carassius gibelio). Furthermore, all these non-native species are known to prefer altered river habitats that resembles more lacustrine environments with low current, well expressed vegetation and eutrophic trends. In general, there are many non-indigeneous fish species which possess completely opposite habitat requirements. A good example could be Oncorhynchus mykiss, which was excluded from this analysis by the authors or Sander lucioperca, Salmo salar, Neogobius fluviatilis. Such species would more represent habitats of a good ecological quality as they prefer running waters rich in oxygen and hard substrates. There are also many indifferent NIFS, which would not present neither bad quality habitats, nor good state habitats. Overall, my suggestion would be to avoid such generalizing statement, that all NIFS could be incorporated as good indicators for assessment of bad water ecological status, but rather talk on possibility to include the particularly assessed four NIFS species into the implementation of ecological quality assessment procedures as indicators of degraded habitats. Overall, it is well known that these four assessed NIFS (Lepomis gibbosus, Pseudorasbora parva, Gambusia holbrooki, and Carassius gibelio) prefers degraded habitats. Thus, particularly these species incorporation into EU Water Framework Directive as biotic indices for ecological health evaluation could be promising.

Reply: We fully appreciate the feedback of the reviewer. In response to your concerns, we have made significant revisions throughout different sections of the main body of the text to provide more specific details about the four non-indigenous species studied (Lepomis gibbosus, Pseudorasbora parva, Gambusia holbrooki, and Carassius gibelio). We acknowledge that these species are known to prefer altered river habitats resembling lacustrine environments with low current, well-expressed vegetation, and eutrophic trends.

In light of your suggestion, we have refined our statements to avoid overgeneralization. We emphasize the potential incorporation of the specifically assessed four NIFS species into ecological quality assessment procedures as indicators of degraded habitats. We agree that caution should be exercised in making broad claims about all NIFS, considering the diverse habitat preferences among these species.

Specific notes:

Discussion chapter:

Line 229: I suggest pointing out the names of particularly assessed non-indigenous fish species rather then to generalized as NIFS. I am certain, that the findings of provided relation trend was very highly depended on the particularly investigated non-indigenous fish species.

Reply: Changed as suggested (lines 268 - 269).

Line 239: Similarly as above suggesting to change NIFS into four particularly fish species Lepomis gibbosus, Pseudorasbora parva, Gambusia holbrooki, and Carassius gibelio.

Reply: Changed as suggested (line 290).

Line 267: Similarly as above suggesting to change NIFS into four particularly fish species Lepomis gibbosus, Pseudorasbora parva, Gambusia holbrooki, and Carassius gibelio.

Reply: Changed as suggested (line 317 - 318).

Reviewer 2 Report

Comments and Suggestions for Authors

This paper is well-written, as an analysis of a dataset which demonstrates a link between presence of non-native fish and habitat quality. The data presented do justify the author’s contention that non-native fish could be used as an indicator of ecosystem health. Although this idea will not be particularly new to environmental managers, the novel element which makes this paper particularly useful is the analysis of data on this scale (115 sites, and multiple indices of habitat quality).

Major suggestion 1: Although references are given to direct the reader to more information about the indices, a few more lines of information need to be given in this paper. This is especially pertinent for the Hellenic Fish Index, for example so that the reader to make better sense of Figure 2 (where density of native fish species is presented in relation to the HEFI). At a minimum, the reader needs to know that the HEFI is derived from four components of the fish community: density of potamodromous species, density of benthic species <15cm, density of omnivorous species <10 cm, and density of insectivorous fish >10cm. This helps the reader to understand why sites can be categorised as ‘poor’ or ‘bad’ using the HEFI despite having a high density of native fishes (Figure 2). Similarly, a brief description of the macroinvertebrate index HESY2 (just one or two sentences) should be given to indicate which feature is being measured (diversity? abundance? biomass?) and for which taxonomic groups. 

Major suggestion 2: Figure 2 and Figure 3 seem to show a correlation with total fish density in total (i.e. native + non-native) and the Moderate, Poor or Bad status on PhQ and HESY2 indices. If that is the case, it should be mentioned, as it appears that index quality affects total fish density (or total fish density affects index quality) regardless of whether the species are native or non-native. The correlation with native species is mentioned in Line 233 – but some explanation as to why native species are also associated with lower ecological quality should be given (if it is a genuine relationship). This is important because the message of the paper is specifically that non-native fish are associated with lower quality.

Major suggestion 3: The introduction and/or discussion could be improved with a few more specific examples from the literature of relationships between degraded habitats and non-native fish assemblages in similar river systems.   

Minor suggested edits:     

Line 18: Add ‘and’ between ‘physicochemical’ and ‘biotic’

Line 48: There are more relevant references that could be cited here relating non-indigenous fish to habitat degradation, some of which relate to Mediterranean-climate river systems, e.g. Colin, N., Villéger, S., Wilkes, M., De Sostoa, A. and Maceda-Veiga, A., 2018. Functional diversity measures revealed impacts of non-native species and habitat degradation on species-poor freshwater fish assemblages. Science of the Total Environment, 625, pp. 861-871. 

Line 76: It would be useful to see a bit more information on the range of habitats covered by these 115 sites – do they include upland tributary streams (where impacts of NIFS are to be less expected) as well as main lowland rivers? Can the reader be pointed towards a map in a reference somewhere?      

Line 78, Line 147, Line 148: The word ‘fish’ should be added after ‘native’ – to make clear that these sentences refer to fish species and not macroinvertebrate species (which are also discussed)

Line 135 and Line 136: The acronym F.O. is not needed here (or should be explained).

Line 136: Use decimal point (not comma) in 34.7

Line 138/139: can any details be given about the sites with a fish community made up only of non-native fishes?

Figure 1 Legend (Line 144): Should read “Percentage of sites with NIFS presence for each quality index and quality status”.

Line 172: The phrase “in contrast” isn’t accurate here. This isn’t really a contrast: mosquitofish and Prussian carp showed higher densities at moderate, poor and bad HESY2 categories and pumpkinseed at bad HESY2 category; and all three species show high densities at poor HEFI.

Figure 6 Legend (Line 188): Mis-spelling of “gibossus”

Figure 8 Legend (Line 219): Should “hydrograph modification” read “hydrological modification” (to match Line 104)?

Line 231: Replace “found to degraded sites” with “found in degraded sites”.

Line 234 to Line 239: It would be good to provide some additional references to support this sentence, as the cited reference is a study covering one indicator and one species (microplastics in Vardar Chub).

Line 259: Insert “a” between “as” and “pressure”

Line 280: Delete “with” in “resemble with other”

Author Response

Reply to Reviewer #2:

This paper is well-written, as an analysis of a dataset which demonstrates a link between presence of non-native fish and habitat quality. The data presented do justify the author’s contention that non-native fish could be used as an indicator of ecosystem health. Although this idea will not be particularly new to environmental managers, the novel element which makes this paper particularly useful is the analysis of data on this scale (115 sites, and multiple indices of habitat quality).

Major suggestion 1: Although references are given to direct the reader to more information about the indices, a few more lines of information need to be given in this paper. This is especially pertinent for the Hellenic Fish Index, for example so that the reader to make better sense of Figure 2 (where density of native fish species is presented in relation to the HEFI). At a minimum, the reader needs to know that the HEFI is derived from four components of the fish community: density of potamodromous species, density of benthic species <15cm, density of omnivorous species <10 cm, and density of insectivorous fish >10cm. This helps the reader to understand why sites can be categorised as ‘poor’ or ‘bad’ using the HEFI despite having a high density of native fishes (Figure 2). Similarly, a brief description of the macroinvertebrate index HESY2 (just one or two sentences) should be given to indicate which feature is being measured (diversity? abundance? biomass?) and for which taxonomic groups.

Reply: We appreciate the reviewer's concerns, and to enhance reader understanding, we have addressed this issue by incorporating additional details into the materials and methods section of the revised text. Specifically, we have provided more information about the metrics used, particularly HEFI and HESY. The relevant information can now be found in a text passage spanning lines 90 to 107.

Major suggestion 2: Figure 2 and Figure 3 seem to show a correlation with total fish density in total (i.e. native + non-native) and the Moderate, Poor or Bad status on PhQ and HESY2 indices. If that is the case, it should be mentioned, as it appears that index quality affects total fish density (or total fish density affects index quality) regardless of whether the species are native or non-native. The correlation with native species is mentioned in Line 233 – but some explanation as to why native species are also associated with lower ecological quality should be given (if it is a genuine relationship). This is important because the message of the paper is specifically that non-native fish are associated with lower quality.

Reply: We appreciate the reviewer's concerns and have made efforts to improve clarity on this matter. Initially, we did not mention the correlation tests between natives and NIFS densities and the ecological quality indices, as the tests did not produce statistically significant results. In response to the reviewer's suggestions, we have now included an explanation about the correlation testing using the Spearman rank correlation in the materials and methods section (lines 133-135). Additionally, we have incorporated information about the outcomes of the results for natives (lines 177-178) and NIFS (lines 1833-184) to provide a more detailed understanding of this aspect.

Major suggestion 3: The introduction and/or discussion could be improved with a few more specific examples from the literature of relationships between degraded habitats and non-native fish assemblages in similar river systems.  

Reply: In line with the reviewer's suggestion, we have enriched the manuscript by incorporating additional examples from the literature concerning the relationships of NIFS, specifically focusing on the Eastern mosquitofish, the Prussian carp, the pumpkinseed sunfish, and the top-mouth gudgeon, with degraded habitats within similar river ecosystems. This expansion has been integrated mostly into the discussion section, contributing to a more comprehensive exploration of the topic. Additionally, the number of references has increased from 37 to 53 to support and strengthen the context of our discussion.

Minor suggested edits:    

Line 18: Add ‘and’ between ‘physicochemical’ and ‘biotic’

Reply: Changed as suggested (line 18).

Line 48: There are more relevant references that could be cited here relating non-indigenous fish to habitat degradation, some of which relate to Mediterranean-climate river systems, e.g. Colin, N., Villéger, S., Wilkes, M., De Sostoa, A. and Maceda-Veiga, A., 2018. Functional diversity measures revealed impacts of non-native species and habitat degradation on species-poor freshwater fish assemblages. Science of the Total Environment, 625, pp. 861-871.

Reply: The reference suggested by the reviewer has been inserted into the text (line 50).

Line 76: It would be useful to see a bit more information on the range of habitats covered by these 115 sites – do they include upland tributary streams (where impacts of NIFS are to be less expected) as well as main lowland rivers? Can the reader be pointed towards a map in a reference somewhere?     

Reply: In response to the reviewer's recommendation and with the aim of enhancing clarity for the reader, we have incorporated a map illustrating the sampling locations utilized in this study. This new information can be found in the text at lines 79 and 108 (Figure 1).

Line 78, Line 147, Line 148: The word ‘fish’ should be added after ‘native’ – to make clear that these sentences refer to fish species and not macroinvertebrate species (which are also discussed)

Reply: Changed as suggested (lines 80, 174, 175).

Line 135 and Line 136: The acronym F.O. is not needed here (or should be explained).

Reply: The text has been altered as suggested by the reviewer (lines 159 -162).

Line 136: Use decimal point (not comma) in 34.7

Reply: Changed as suggested (line 161).

Line 138/139: can any details be given about the sites with a fish community made up only of non-native fishes?

Reply: The inclusion of the map provides additional details about the location of sampling sites exclusively with NIFS (See Figure 1).

Figure 1 Legend (Line 144): Should read “Percentage of sites with NIFS presence for each quality index and quality status”.

Reply: Changed as suggested (line 170).

Line 172: The phrase “in contrast” isn’t accurate here. This isn’t really a contrast: mosquitofish and Prussian carp showed higher densities at moderate, poor and bad HESY2 categories and pumpkinseed at bad HESY2 category; and all three species show high densities at poor HEFI.

Reply: Changed as suggested. (line 201).

Figure 6 Legend (Line 188): Misspelling of “gibossus”

Reply: Changed as suggested (line 217).

Figure 8 Legend (Line 219): Should “hydrograph modification” read “hydrological modification” (to match Line 104)?

Reply: Changed as suggested (line 248).

Line 231: Replace “found to degraded sites” with “found in degraded sites”.

Reply: Changed as suggested (line 270).

Line 234 to Line 239: It would be good to provide some additional references to support this sentence, as the cited reference is a study covering one indicator and one species (microplastics in Vardar Chub).

Reply: More references were given (lines 289).

Line 259: Insert “a” between “as” and “pressure”

Reply: Changed as suggested (line 309).

Line 280: Delete “with” in “resemble with other”

Reply: Deleted as suggested by the reviewer (line 334).

Reviewer 3 Report

Comments and Suggestions for Authors

There are no references to the known physiology of some of the introduced fishes, such as Gambusia about which there is a lot published about its resistance to poooor water quality. On the otehr hand, sunfishes are not known (?) to be so tolertant about wtaer quality. I do not think you can lump all the introdcued fishes into the same category in terms of water quality tolerances. Further how to the introdcued fishes relate taxonmically to any dsplaced native fishes? are any eco-analogues? 

Author Response

Reply to Reviewer #3:

There are no references to the known physiology of some of the introduced fishes, such as Gambusia about which there is a lot published about its resistance to poooor water quality. On the otehr hand, sunfishes are not known (?) to be so tolertant about wtaer quality. I do not think you can lump all the introdcued fishes into the same category in terms of water quality tolerances. Further how to the introdcued fishes relate taxonmically to any dsplaced native fishes? are any eco-analogues?

Reply: Followint the reviewers concerns we have now taken into account

Reply: We appreciate the reviewer's concerns, and we have carefully considered your doubts, making substantial enhancements to the text. In particular, we have incorporated additional references to provide examples that specifically highlight the known physiology of the four NIFS, with specific attention to their tolerance to water quality. Furthermore, in response to your input, we have refined our statements to steer clear of broad generalizations. We have also addressed the relationship between the introduced fish and native fishes, focusing on this aspect, especially in the discussion section.